# The Predictive Power of the User Cost Spread for Economic Recession in China and the US

**Dongfeng Chang [1], Ryan S. Mattson [2] and Biyan Tang [3,*]**

[1]   School of Economics, Shandong University, Jinan 250100, Shandong, China; dchang@sdu.edu.cn
[2]   The Center for Financial Stability, New York, NY 10036, USA; rmattson@the-cfs.org
[3]   University of Massachusetts Dartmouth, Economics Department, North Dartmouth, MA 02747-2300, USA
[*]   Correspondence: btang@umassd.edu

**Abstract:** The predictive power of the yield curve slope, or the yield spread is well established in the United States (US) and European Union (EU) countries since 1998. However, there exists a gap in the literature on the predictive power of the yield spread on the Chinese economy. This paper provides a different leading recession indicator using the Chinese and US economy as comparative examples: the user cost spread, being the difference of the opportunity costs of holding government securities of different maturities. We argue that the user cost spread, based on the Divisia monetary aggregate data like the ones produced by the Center for Financial Stability, provides improved predictive ability and a better intuitive explanation based on changes in the user cost price of holding bonds.

**Keywords:** Divisia monetary aggregates; user cost spread; recession; China; yield spread

## 1. Introduction

When returns on long maturity bonds fall below those of shorter-term bonds, the yield spread indicator becomes negative. This usually heralds a recession within one year, as detailed in Harvey (1988) and Estrella and Trubin (2006). To understand this infamous prediction, the components of longer-term bond interest rates must be explained. Potential reasons abound, for example, the liquidity premium theory, stating that long-term bonds and short-term bonds are substitutes, as proposed by Tobin (1969) and its succeeding line of literature. The average interest rate on longer maturity bonds is equal to the average of the current short-term maturity bonds and the expected short-term returns over the lifespan of the longer maturity plus a liquidity premium. By this reasoning, when the yield spread drops below zero, the following occurs:

- Markets expect future short-term rates to be lower than present ones, presumably due to an expansionary policy by the central bank in response to weaker economic growth;
- There exists a smaller liquidity premium, or, in other words, investors may need less compensation for holding on to longer-term maturities.

A lower liquidity premium reflects expectations of lower future inflation, or growing demand for safer longer-term securities. A higher demand for longer maturity bonds will result in a higher price, corresponding to lower longer-term returns. More generally, we can expect the yield spread to go negative when a central bank switches from a policy of monetary contraction to a policy of monetary expansion, which tends to happen as the growth in an economy slows and a recession potentially begins.

The recession predictive property of the yield spread is well established starting with the seminal work of Harvey (1988) formalizing the observed relationship. Empirical support was developed in Estrella and Mishkin (1996), Estrella and Trubin (2006), and Rudebusch and Williams (2009). Wheelock and Wohar (2009) listed all relevant papers in an exhaustive survey on the relationship between yield spreads and output growth. Expanding on the foundation of Tobin (1969), Marzo and Zagaglia (2018) and Canzoneri et al. (2011) provided a convincing theoretical groundwork for this relationship in transaction costs and liquidity theory that is not dissimilar to our approach using the user cost price of holding on to less liquid assets. Importantly, this approach is similar to Mattson (2019), which provided a short economic intuition and empirical result showing that the relationship can be explained by the liquidity tradeoff between the service of bonds as a store of value and medium of exchange (in asset backed securities, for example) using the Divisia monetary aggregate literature beginning with Barnett (1978, 1980)—specifically, the construction of the user cost of the durable service provided by money[1] .

While this literature extensively covers the United States (US) and European cases, there is a dearth of literature on bond yields and recessions in China and other emerging economies, despite the growing importance of bond markets in that context. The notable exceptions are papers that attempted to estimate the slope and curvature of the yield curve, such as Yan and Guo (2016).

This paper begins to fill that gap for the Chinese case and provides reasoning behind the predictive property of a segmented market and liquidity premium intuition through the Divisia cost interpretation. Section 2 of this paper provides the reasoning for the yield curve predictive power, as well as an explanation of the literature on the user cost spread. Section 3 describes the empirical methodology used to evaluate the predictive power of the yield spread and the user cost spread. Section 4 describes and interprets the results, and leads into the conclusion in Section 5.

## 2. Previous Literature

### 2.1. Estimation and Intuition behind the Yield Spread Prediction

Since Harvey (1988), the yield spread is used in predicting changes in output, industrial production, or the occurrence of a recession. Harvey (1988) claimed that expected real-term structure contains more information than lagged consumption growth and lagged stock returns when used for forecasting economic growth. Estrella and Estrella and Hardouvelis (1991) first popularized the treasury term spread as a significant and leading predictor of future output growth and recession. Estrella and Mishkin (1996) examined the performance of the yield curve spread and other macroeconomic variables in predicting downturns in the business cycle, leading to the conclusion shared by the simplified case of Estrella and Trubin (2006) that the yield spread outperforms other indicators in a one year horizon. In these cases, the Probit model was used with the 10-year and three-month treasury spread. It is noted in this work, however, that the recession probability levels dropped overall in the 1990s and 2000s.

Extensions exist on this simplified Probit model approach. Chauvet and Potter (2002, 2005) studied the effects of structural break points on the probability of recession from Probit models, and concluded that models with break points and autocorrelated errors fit better in sample than the basic Probit prediction for one year out. Rudebusch and Williams (2009) explained the puzzle of why the univariate Probit using the yield spread outperforms the Survey of Professional Forecasters, due to the lack of weight given to bond rates in their models. Wheelock and Wohar (2009) extensively surveyed the literature in prediction of recessions and output growth in the 1980s and 1990s. International studies on the yield spread also support its forecasting ability. For example, Duatre et al. (2005) and Zagaglia (2013) ran estimations for the European Union. Nyberg (2010) examined financial variables as predictors of the probability of recession in Germany while proposing a new dynamic Probit model with better performance than the static model presented by Estrella and Trubin (2006).

---

[1]    The literature on the user cost of durable goods and services begins with Diewert (1976).

Liu and Moench (2016) reassessed the ins and outs of sample predictability of US recessions for a large number of leading indicator variables, using the treasury term spread as a benchmark. Both univariate and multivariate Probit models were used to evaluate the relative model performance based on the receiver operating characteristic (ROC) curve. At shorter horizons, other predicting indicators improved the recession forecast precision significantly compared to the term spread, especially for the three- and six-month-ahead horizon, with the annual return on the Standard and Poor's 500 equity index (S&P500) providing the strongest improvement. At longer horizons, the treasury term spread was more difficult to outperform.

China has less available research for the yield spread despite a growing importance of bond markets in emerging economies. Mehl (2009) examined the forecasting power of the slope of the yield curve within certain emerging economies and concluded that the spread contains information for future inflation and output growth, with differences across countries linked to market liquidity, in line with the conclusions in Canzoneri et al. (2011) and the Divisia literature from Barnett and Serletis (2005) for monetary aggregates. Yan and Guo (2016) used a dynamic Nelson–Siegel model to estimate the level, slope, and curvature of the yield curve, then utilized a vector autoregression to test its correlation to macroeconomic indicators. There was no significant link found in yield curve adjustments and these indicators. The focus of the current literature remains on the estimation of the shape and curvature of the yield curve, but not in econometrically forecasting the probability of a recession in emerging markets.

### 2.2. The User Cost Spread and the Divisia Literature Interpretation

An established literature provides a variety of potential explanations as described earlier, such as the liquidity premium theory, segmented markets, and the expectations hypothesis. The latter two point out the key fact that different maturities of different bonds behave as near substitutes, but this is not explicitly considered by observing only interest rates. The pricing of these bonds as a "durable service" of "store of value" and "medium of exchange", as is done with monetary assets in the Divisia literature, allows for a simplified price-dependent explanation through supply and demand of substitute goods.

We re-interpret the yield spread as an opportunity cost difference of holding long maturity store of value asset services over a shorter-term medium of exchange (liquid) asset as described in Barnett (1978, 1980) to develop the proper pricing of the near substitute monetary assets like checking accounts, savings accounts, and money market mutual funds, all of which hold different interest rates and, therefore, different prices to hold. The "user cost spread", which demonstrates the current relative price difference between the store of value services of bonds, will provide the motivation for the recession prediction.

In the wake of the 2008 financial crisis and succeeding Great Recession, short-term yields pushed down to the zero lower bound. The lack of information produced by the Federal Reserve's preferred intermediate targets for monetary policy left ambiguity regarding the effectiveness of lower short-term interest rates as indicators of monetary easing, while Divisia monetary aggregates demonstrated a clear contraction throughout 2009 and into 2010 (see Barnett et al. (2012) and Belongia and Ireland (2015)). A rethinking and refocus on the aggregates and their user costs produces a new perspective and explanation for the macroeconomic trends; for example, Mattson and Valcarcel (2016) demonstrated a significant and prolonged user cost compression of liquid and non-liquid monetary assets.

We can directly establish the link between the yield spread and the user cost pricing of the Divisia monetary literature. The Divisia user cost lines up theoretically with money being a durable service in the same way bonds provide "store of value" service and, in some cases, liquidity when packaged within asset-backed securities. Therefore, the price of holding these bonds of different maturities is defined as their user cost given the asset services they provide. The user cost difference is trivially defined as the yield spread adjusted for the distance to the return on some benchmark pure store of value asset. Within the Divisia literature and assets on the pricing of durables by Diewert (1976),

all monetary assets are priced by the opportunity cost of holding them. Intuitively, the user cost proof from Barnett (1978, 1980) was extended to multiple assets not clustered as monetary assets and considered to be highly illiquid, although still more liquid than the pure store of value benchmark asset. In that case, the user cost price of any bond with $i$ maturity, at time $t$, with a return of $r_{i,t}$ compared to a benchmark rate $R_t$ of the pure store of value asset is

$$\pi_{i,t} = \frac{R_t - r_{i,t}}{100 + R_t}, \tag{1}$$

or simply the normalized difference of the return on the individual bond relative to a benchmark rate of return on a pure store of value service. If we then consider some other longer maturity bond $j > i$, the difference of the two user costs is then

$$\pi_{i,t} - \pi_{j,t} = \frac{(R_t - r_{i,t}) - (R_t - r_{j,t})}{100 + R_t}, \tag{2}$$

simplifying to

$$\widetilde{\pi}_t = \pi_{i,t} - \pi_{j,t} = \frac{r_{j,t} - r_{i,t}}{100 + R_t}. \tag{3}$$

Thus, the yield spread is a naïve user cost difference which is not normalized to the benchmark rate of return within the economy. While the two will provide similar empirical outcomes, the advantage of a Divisia user cost interpretation lies in the straightforward definition of the prices of holding these bonds versus couching the potential explanation for their predictive power in terms of expected returns.

This approach using the Divisia method of pricing incorporates the segmented markets hypothesis by treating bonds of different maturities as imperfect substitutes. Indeed, one of the main contributions of the Divisia monetary aggregate literature is to uncover and acknowledge the failings of the simple sum approach that treats all monetary assets as perfect substitutes. Furthermore, the expectations of the interest rates in this case can be put aside as all bonds are treated as imperfect substitutes based on their liquidity and store of value pricing at the present time period, although a more complex expectations hypothesis could be developed as demonstrated in the monetary asset case in Barnett and Wu (2005).

## 3. Methodology and Data

The sources for both US and China data are described in Section 3.1. For most of the US data, the yield spread data are from Federal Reserve Economic Data (FRED), the recession index is from the Organization for Economic Cooperation and Development (OECD), and the benchmark rate for the US, used to compute the user-cost spread, is from the Center for Financial Stability (CFS). Table 1 describes the relevant Chinese data sources.

**Table 1.** Data sources for China from January 2002 to August 2018.

| Time Series | Data Source |
| --- | --- |
| 10-year government bond yield | China Central Depository & Clearing Co., Ltd. (CCDC) |
| 2-year government bond yield | China Central Depository & Clearing Co., Ltd. (CCDC) |
| Benchmark rate, 1-year bank loan rate | The People's Bank of China |
| OECD Recession Index | OECD and FRED |
| OECD Leading Index | OECD and FRED |
| CHIBOR | The People's Bank of China |

In this paper, we employ the traditional Probit model, the previous literature's choice of model. The Probit model is incorporated into most of the yield spread literature; therefore, we in this paper follow that tradition, while recognizing that there are improvements in both Logit and Probit estimation that could provide a fruitful extension to this research agenda.

### 3.1. Data Sources

All data in this paper were taken from public resources. Firstly, the recession data came from the OECD-based recession indicators. The indicator is 1 when the economy is in a recession and 0 otherwise. The yield spread between the 10-year maturity bond and the two-year bond rate for the US was taken from the St. Louis Fed's Federal Reserve Economic Data tool (FRED). The unique identifying tags for these series are DGS10 and DGS2, respectively. The US user cost spread was calculated using the lending benchmark rate provided by the Center for Financial Stability in their Advances in Monetary and Financial Measurement data set and the Chinese Interbank Loan Rate (CHIBOR). The OECD leading index is composed of several time series, all available from the OECD datasets online. The time period considered for the US economy was from January 1968 to September 2018, while, in the Chinese case, it was from January 2002 to August 2018 due to availability of data.

### 3.2. The Probit Recession Prediction Model

Since the recession indicator is a binary variable, with discrete observations of 0 for "no recession" and 1 for "recession", and since the yield spread is a continuous variable observed as negative or positive, the proper method for estimation is binary-dependent variable regression. While it is possible to use standard ordinary least squares in the "linear probability model", this can lead to estimated probabilities below zero and above one. The interpretation of these kinds of figures becomes difficult with those unreasonable probabilities. Therefore, the Logit and Probit models are most often used to constrain the estimation to some forecasted probability between zero and one. Simplified Probit models are used to forecast the recession probability in the next $h$ period with the information available at time $t$. As in Estrella and Trubin (2006), we consider the probability of a recession a year in advance; thus, in dealing with our monthly data, $h = 12$. The most naïve form of our models is the univariate Probit model relating the yield spread to the probability of a recession in a year.

$$P\big(OECD_{t,t+h} = 1\big) = F(\alpha_0 + \alpha_1 s_t), \tag{4}$$

where $s_t$ denotes the spread of the longer-term and shorter-term maturities, while $OECD_{t,t+h}$ denotes the value 1 if there is a recession for $h$ periods ahead and 0 if not. We define for the Probit model the cumulative normal distribution function,

$$F(z) = \int_{-\infty}^{z} \frac{1}{\sqrt{2\pi}} \exp\left(\frac{-x^2}{2}\right) dx. \tag{5}$$

We compare this model from Equation (4) to a similar naïve form using the user cost spread as defined in Equation (3) before

$$P\big(OECD_{t,t+h} = 1\big) = F(\beta_0 + \beta_1 \widetilde{\pi}_t). \tag{6}$$

We can expand these models to include the leading OECD indicator, $\ell^u$ and $\ell^c$, as well as the target short-term rate of the Federal Funds Rate in the US and CHIBOR for China, denoted by $i_t$.

$$P\big(OECD_{t,t+h} = 1\big) = F(\theta_0 + \theta \widetilde{\pi}_t + \theta_2 \ell_t); \tag{7}$$

$$P\big(OECD_{t,t+h} = 1\big) = F(\theta_0 + \theta \widetilde{\pi}_t + \theta_2 \ell_t + \theta_3 i_t). \tag{8}$$

We compare for both the US and China the performance of the user cost spread relative to the traditional yield spread. In each case, we find either marginal improvement or comparable results to the yield spread; thus, using the user cost spread may improve the forecast, but will not make it worse. In terms of intuition, the user cost spread provides the results presented in the next section, with the in-and-out sample forecast and a comparison of goodness of fit and information criterion measures for the Probit model.

## 4. Results

The two tables below demonstrate the regression results of each of the four models for both US data and Chinese data. The first column compares the univariate Probit model, while the next columns includes the OECD Leading Indicator Index, and finally the last two columns look at the full models from Equations (7) and (8) that also include the overnight interest rate. Table 2 focuses on the Chinese data and estimations that compare the user cost spread predictive power with the yield spread prediction.

**Table 2.** Probit model results for China, January 2002 to August 2018.

| Model | (1) | (2) | (3) | (4) | (5) | (6) |
|---|---|---|---|---|---|---|
| Yield spread | −1.239 *** | −1.783 *** | −1.898 *** | | | |
| | (0.237) | (0.295) | (0.372) | | | |
| Leading indicator | | −0.566 *** | −0.561 *** | | −0.569 *** | −0.563 *** |
| | | (0.099) | (0.099) | | (0.100) | (0.100) |
| CHIBOR | | | −0.099 | | | −0.109 |
| | | | (0.189) | | | (0.190) |
| User cost spread | | | | −131.037 *** | −189.270 *** | −202.643 *** |
| | | | | (25.041) | (31.235) | (39.566) |
| Constant | 0.585 *** | 57.745 *** | 57.508 *** | 0.586 *** | 58.072 *** | 57.834 *** |
| | (0.196) | (10.034) | (10.013) | (0.196) | (10.076) | (10.052) |
| McFadden $R^2$ | 0.1257 | 0.3240 | 0.3251 | 0.1262 | 0.3256 | 0.3268 |
| In-sample MAE | 0.2833 | 0.1656 | 0.1637 | 0.2823 | 0.1640 | 0.1623 |
| In-sample RMSE | 0.3659 | 0.2924 | 0.2915 | 0.3649 | 0.2910 | 0.2903 |
| Out-of-sample MAE | 0.5475 | 0.4851 | 0.4990 | 0.5484 | 0.4854 | 0.4988 |
| Out-of-sample RMSE | 0.5680 | 0.5751 | 0.5823 | 0.5686 | 0.5753 | 0.5824 |
| $N$ | 200 | 200 | 200 | 200 | 200 | 200 |
| AIC | 232.516 | 182.677 | 184.405 | 232.378 | 182.268 | 183.944 |

Note: $p < 0.01$ ***.

From Table 2, we can see that the user cost spread marginally predicts a recession in China. Quantitatively, the pseudo-$R^2$ for the models that use the user cost spread instead of the interest rate spread is higher, demonstrating less error based on the normalization of the distance to the benchmark rate. The results are mixed for the mean absolute error (MAE) and root-mean-squared error (RMSE), which provide a lower in-sample but not out-of-sample error for the user cost spread relative to the yield spread. This result differs from the marginal improvement in the US recession predictions in Mattson (2019) and the results of this paper for the US in Table 3. The addition of the leading index improves the in-sample and out-of-sample forecast, as well as the goodness of fit according to the McFadden $R^2$; however, the CHIBOR rate is not significant in predicting the recession when included with the spreads and the leading index.

Figure 1 shows us the in-sample predicted recession probabilities for the Chinese economy based on the user-cost spread. When the recession probabilities hit around or above 50%, the Chinese economy will end up in a recession, for example, the 2008 recession, with 2012 slowing down and 2014 slowing down.

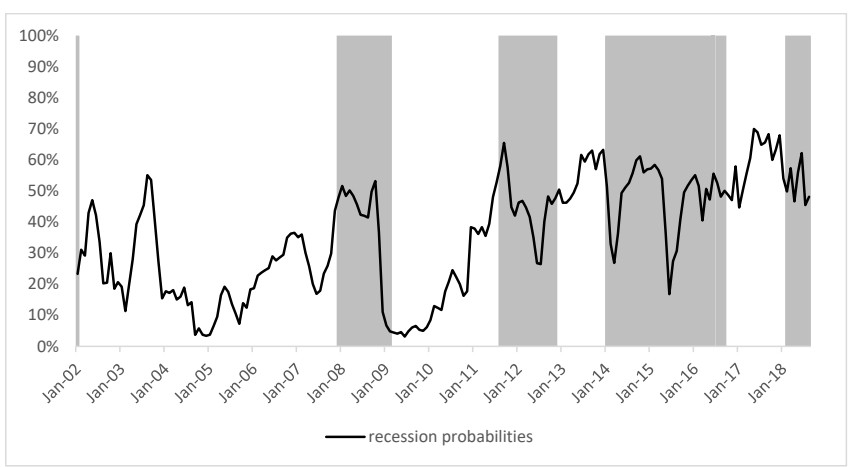

**Figure 1.** China probability of recession using the user cost spread.

From Figure 2, we can conclude that the probabilities of the Chinese economy going into recession approach 70% when the user cost spread nears zero, and it reaches almost 45% when the user cost spread is about 0.005. As the user cost spread becomes more positive, the recession probability of the Chinese economy decreases dramatically, and it decreases to 10% when the user cost spread reaches 0.015.

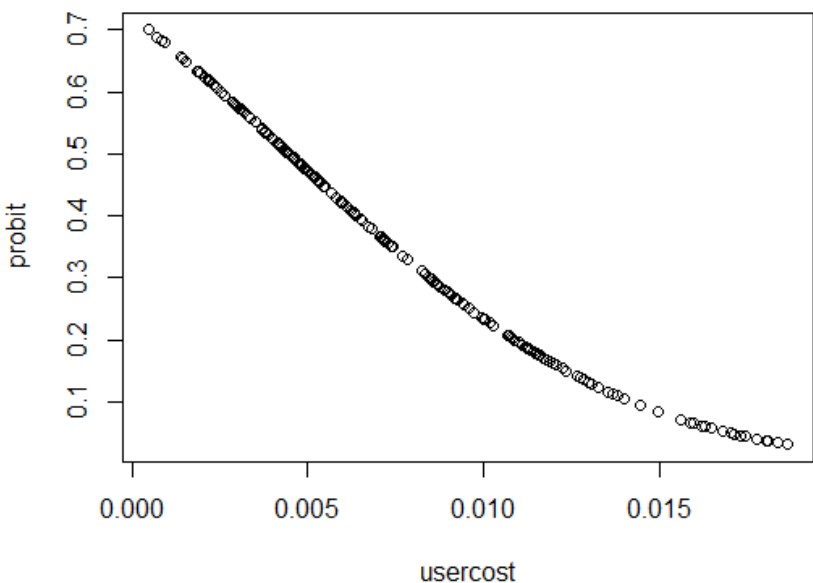

**Figure 2.** The Probit recession prediction and the Chinese user cost spread.

**Table 3.** Probit model results for the United States, January 1968 to August 2018.

| Model | (1) | (2) | (3) | (4) | (5) | (6) |
|---|---|---|---|---|---|---|
| Yield spread | −0.742 *** | −0.627 *** | −0.606 *** | | | |
| | (0.075) | (0.089) | (0.372) | | | |
| Leading indicator | | −0.702 *** | −0.702 *** | | −0.702 *** | −0.703 *** |
| | | (0.078) | (0.078) | | (0.078) | (0.078) |
| Fed funds | | | 0.012 | | | −0.007 |
| | | | (0.025) | | | (0.025) |
| User cost spread | | | | −81.072 *** | −69.141 *** | −97.702 *** |
| | | | | (8.153) | (9.744) | (11.282) |
| Constant | −0.303 *** | 69.176 *** | 69.117 *** | −0.281 *** | 69.284 *** | 69.253 *** |
| | (0.096) | (7.719) | (7.715) | (0.096) | (7.718) | (7.717) |
| McFadden $R^2$ | 0.3111 | 0.5811 | 0.8510 | 0.3163 | 0.5862 | 0.5864 |
| In-sample MAE | 0.2042 | 0.0249 | 0.0250 | 0.2039 | 0.1176 | 0.1177 |
| In-sample RMSE | 0.3200 | 0.2411 | 0.2411 | 0.3192 | 0.2397 | 0.2398 |
| Out-of-sample MAE | 0.0430 | 0.0249 | 0.0250 | 0.0364 | 0.0213 | 0.0207 |
| Out-of-sample RMSE | 0.0562 | 0.0400 | 0.0401 | 0.0494 | 0.0346 | 0.0354 |
| N | 609 | 609 | 609 | 609 | 609 | 609 |
| AIC | 338.093 | 209.401 | 211.162 | 335.553 | 206.693 | 208.598 |

Note: $p < 0.01$ ***.

From Table 3, we see that the user cost spread can marginally predict the US recession better, which can be seen from the pseudo-$R^2$, the root-mean-squared error (RMSE), and the mean absolute error (MAE). This result is consistent with the Mattson (2019) paper's conclusion. Also, the OECD leading index is 0.001 significant in predicting US recession, and the average effective Federal Fund rate is not significant, which maybe can be explained by the fact that the Federal Funds Rate information does not change that much, especially during the out-of-sample period for the US.

Overall, based on the results from US and Chinese regressions, we can claim that the user cost spread can marginally predict the US and Chinese recession better as measured from the values of pseudo-$R^2$, the root-mean-squared error (RMSE), and the mean absolute error (MAE). In the US, the pseudo-$R^2$ for the user cost spread model is 0.3163441 compared to 0.3111071 for the yield spread model, a 1.7% improvement in terms of explaining the variance of the data. Another tool that is typically used to measure the accuracy of a model's predicating power is the root-mean-squared error (RMSE) or the mean absolute error (MAE), where the lower the error is, the more accurate the prediction will be. Both the in-sample and out-of-sample RMSEs for the US economy from the traditional yield spread model are higher than those of the user cost spread models, which means that the yield spread model produces less accurate results than the user cost spread models. Specifically, the user cost spread model improved the accuracy of the predicting power in terms of the out-of-sample RMSE and MAE by 12.04% and 15.31% for the US.

For China, the improvement of the pseudo-$R^2$ of the user cost spread model is smaller at 0.45%. Moreover, for the in-sample RMSE and MAE, the improvements of model 2 are 0.28% and 0.38%. However, the out-of-sample RMSE and MAE for model 2 indicate less accurate predictions compared to model 1, although the change is small. Overall, the trend is that model 2 produces better results. The possible reason is that, during this time span of Jan 2002 to August 2018, China's economy had a regime change as explained in Barnett and Tang's (2016) paper. The magnitude of improvement is much more significant due to the US data's long history, and a more mature financial market than that of China.

From Figure 3, we can conclude that the user-cost spread and the predicted U.S. recession probabilities from the Probit model have a negative relationship. As the user-cost spread reaches negative 0.02, the recession probability almost reaches 90%. When the user cost spared is -0.01, the recession probabilities for U.S. economy reaches 70%, and as the user-cost spread reaches 0.02, the recession probabilities is as low as almost 0%. This graph shows a very clear trend; when the user-cost spread become negative the chance of U.S. economy turns to recession increase quickly, and the probability reaches almost 100% when the spread become −0.015.

## usercost prediction of recession for US

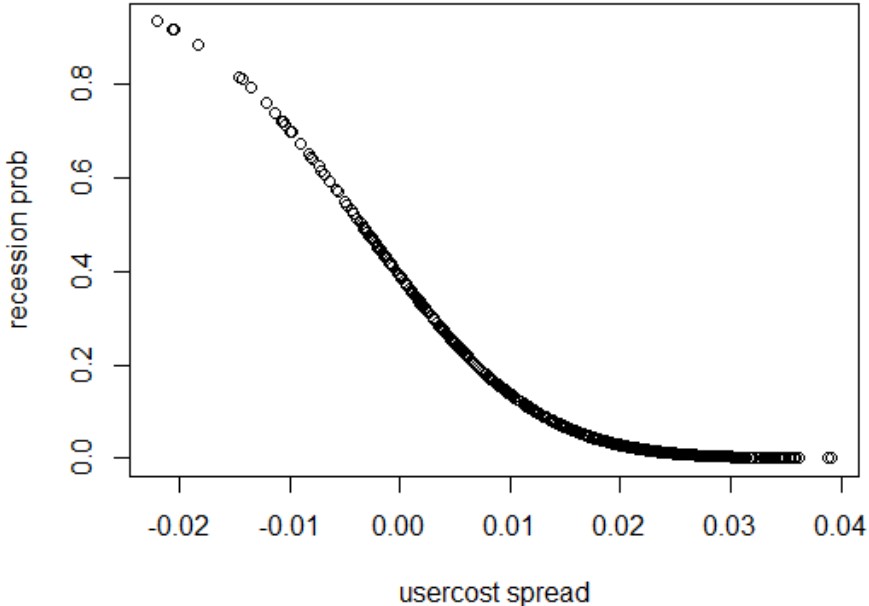

**Figure 3.** United States (US) recession probability forecast with the user cost spread.

Figure 4 depicts the out-of-sample recession probabilities for the U.S. from September 2009 to September 2018 using the user-cost spread as explanatory variable. It shows a very low probabilities in general for this period. The highest recession probability is about 13% for September 18, and the lowest recession probabilities is almost 0% from May 2010 to May 2011. These low forecasted recession probabilities are due to the fact that the U.S. economy has been expanding for this period. Figure 5 demonstrates the in-sample forecasted recession probabilities for the U. S. from Jan 1968 to Sep 2018 using user-cost spread as the independent variable. This figure shows that for the past seven U.S. recessions during this period, the user-cost spread has successfully predicted them. For the 1981–1982 recession, the predicted probability reached as high as 91.7%. And most of these seven recession probabilities were above 50%. Even for the least accurate prediction of 1990 recession, the probability reached 33.1%, with before and after recession period recession probabilities being extremely low, which means the 33% of recession chances was a clear and strong warning signal. Therefore, overall, the user-cost spread is a very good indicator to forecast the recession within US economy from Jan 1968 to 2018.

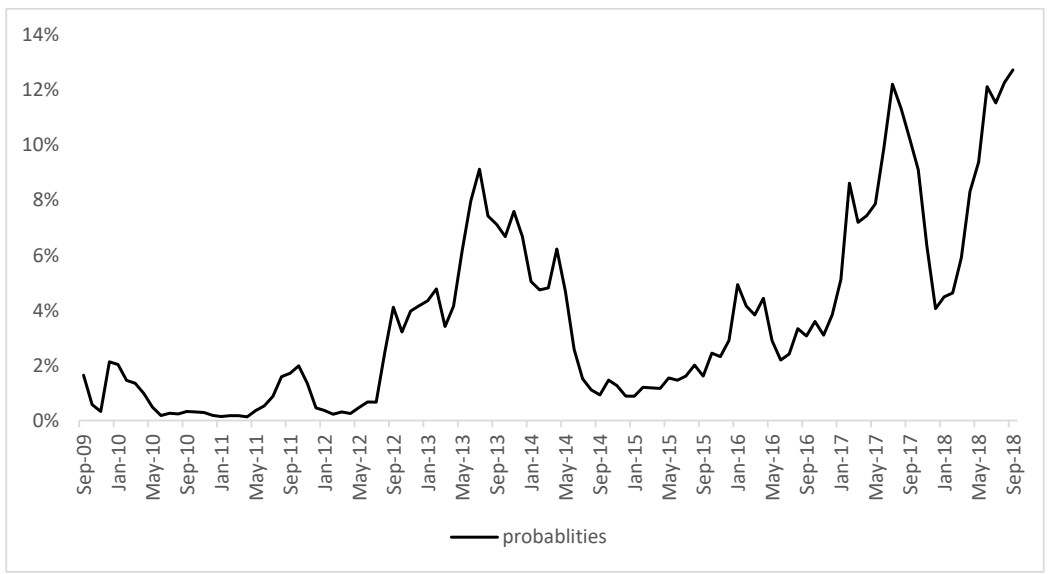

**Figure 4.** Out-of-sample user cost forecast recession probabilities for the US: September 2009 to September 2018.

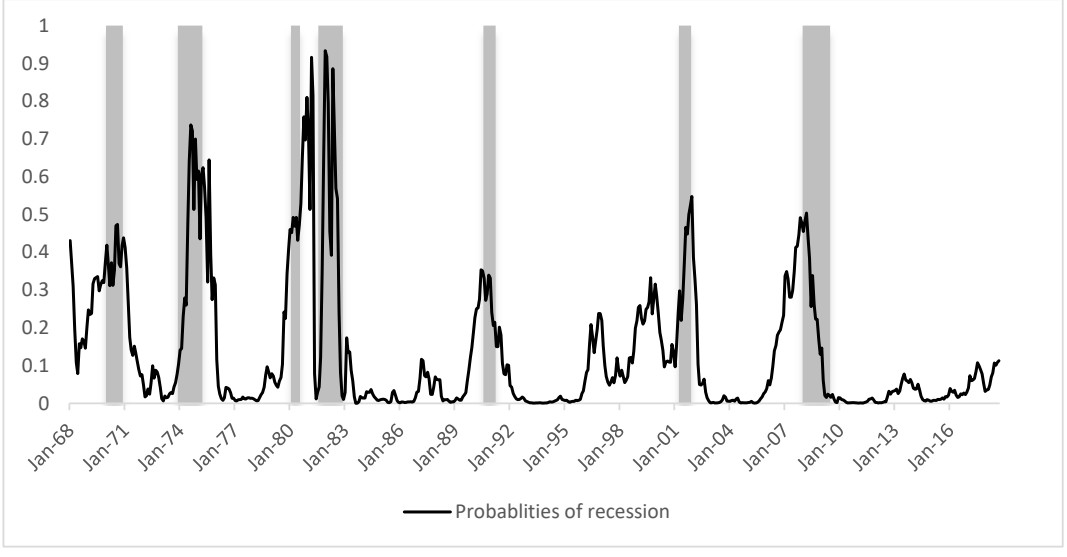

**Figure 5.** User cost forecast recession probabilities for the U.S. from January 1968 to September 2018.

As shown in Table 4, for both the US and China data, we can see that user cost spread can marginally predict the economic recession better, and, by adding the OECD based leading index, the accuracy increased tremendously, whereby the $R^2$ almost doubled for the US case, and it almost tripled for the Chinese case. The user cost spread and the interest rate spread improve the recession forecasts for China, implying a potentially rich econometric literature to be tapped in the same way the yield spread worked for the US and the EU. However, the interpretation of these results remains uncertain: how can one best explain the reason for the inversion before a recession? The segmented markets and expectations hypotheses provide credible alternatives, but the liquidity preference framework has a ready interpretation through the user cost price of holding onto longer-term assets. As the user cost difference between holding a short-term bond as store of value and medium of exchange relative to holding a long-term bond for the same reason drops below zero, the market is preferring short-term liquidity: they expect a slow-down and a recession. In fact, the bond market behavior signals well in advance the slow-down as demand for the substitute short-term bonds increases relative to the long-term bonds and the price difference compresses. A similar behavior can be seen in the analysis of Mattson and Valcarcel (2016) on monetary assets during the great compression of user costs; as the consumers began preferring liquidity to store of value in the face of dropping interest rates, the user cost differences between liquid "medium of exchange" assets like cash compressed to those "store of value" assets like money market mutual funds and commercial paper. The bond market behaves similarly as bonds of different maturities behave as substitutes to each other, in line with the liquidity preference theory of Tobin (1969), the user cost pricing of Diewert (1976), and the Divisia aggregation literature of Barnett (1980).

**Table 4.** Model 2 of user cost spread improvements compared to Model 1 of yield spread.

| United States | Yield Spread/Model 1 | User Cost Spread/Model 2 | Improvement (%) | Interpretation |
|---|---|---|---|---|
| Pseudo $R^2$ | 0.3111071 | 0.3163441 | 1.68 | More variance explained |
| Out-of-sample RMSE | 0.05618087 | 0.04941532 | −12.04 | More accurate |
| In-sample RMSE | 0.3199803 | 0.3192404 | −0.23 | More accurate |
| Out-of-sample MAE | 0.04299932 | 0.03641298 | −15.32 | More accurate |
| In-sample MAE | 0.2042593 | 0.203881 | −0.19 | More accurate |
| **China** | **Yield Spread/Model 1** | **User Cost Spread/Model 2** | **Improvement (%)** | **Interpretation** |
| Pseudo $R^2$ | 0.1256899 | 0.1262178 | 0.42 | More variance explained |
| Out-of-sample RMSE | 0.568027 | 0.5686311 | 0.11 | Less accurate |
| In-sample RMSE | 0.365927 | 0.364891 | −0.28 | More accurate |
| Out-of-sample MAE | 0.5474695 | 0.5484019 | 0.17 | Less accurate |
| In-sample MAE | 0.2832841 | 0.2822546 | −0.36 | More accurate |

## 5. Conclusions

The gains in prediction for interpreting the yield spread effects through the lens of a user cost difference are marginal in the in-sample and out-of-sample forecasts. However, they do not worsen the recession predictions and provide a unique interpretation for the link of the yield spread and recessions through the cost of holding short- versus long-term bonds, as fleshed out in Section 2 of this paper and Mattson (2019). We further filled a gap in the literature regarding recession prediction and yield spreads in China through recent and publicly available data, and further improved those predictions using the OECD leading index. We did not find much significance or use for including the short-term loan rates in these predictions.

Overall, from both the US and China data, we can see that user cost spread can marginally predict the economic recession with more fit based on the McFadden $R^2$ measure. By adding the OECD based leading index, the accuracy increased tremendously, whereby the $R^2$ almost doubled for the US case, and it almost tripled for the Chinese case.

We do not, however, delve into the reasons for using Probit over Logit, nor do we include the nuances of regime change by the central bank in either country. As one potential extension, it was shown in Train (2003) and Tsagkanos (2007) that a mixed Logit approach improves analysis for emerging markets in a discrete binary variable environment. Such an approach could provide more evidence for or against the use of the user cost spread in lieu of the yield spread, given the distance to the benchmark rate. For further research, we will look into the structure break in the economy's impact on the predicting power of yield spread or user cost spread, and explore other leading indexes, such as the S&P 500 index annual rate of turn, or the corresponding Chinese stock market rate of return, as well as manufacturing activity, etc.

Our results of user cost spread's better predicting power of future economic recession shed light on macroeconomics policy decisions. As the results show, the user cost spread can predict recession much better than the yield spread, and the monetary policy decision-makers can rely on the user cost spread for future policy directions, especially in an environment where low interest is the norm, and the yield spread gradually loses its power of recession prediction. The policy implications for China are two-fold. Firstly, the user cost and user cost spread are useful economic indicators that contain reliable information for Chinese economy conditions. For example, when the user cost of money increases, this can signal to the monetary authority that the borrowing cost is too high for the firms to efficiently invest. User cost spread can be a useful recession index for policy-makers and general investors. Secondly, as the results show, China's regression models produce smaller pseudo-$R^2$ values and bigger RMSEs and MAEs. This can be explained by, on the one hand, the short history of economic data, and, on the other hand, the still developing yet immature financial market, or the dual track bond markets of China. Thus, the implication is that China can benefit more from developing the bond market and financial market as described in Tsagkanos et al.'s (2019) paper on developing the Greek financial market to facilitate foreign direct investment in Greece. This can improve the efficiency of the market, allocating the financial resources to more productive and efficient industries.

**Author Contributions:** Conceptualization, B.T., R.S.M. and D.C.; Data curation, B.T. and R.S.M.; Formal analysis, B.T. and R.S.M.; Funding acquisition, D.C.; Methodology, B.T., R.S.M. and D.C.; Software, B.T., R.S.M. and D.C.; Validation, D.C.; Writing—original draft, B.T. and R.S.M.; Writing—review & editing, B.T., R.S.M. and D.C.

**Funding:** This research received no external funding.

**Acknowledgments:** We are indebted to the two anonymous referees for their invaluable contributions to the paper. We further wish to thank John Francois for his valuable input and the students at West Texas A&M's Monetary Theory Seminar for their contributions.

**Conflicts of Interest:** The authors declare no conflicts of interest.

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
