# Peer review of "The Predictive Power of the User Cost Spread for Economic Recession in China and the US"

_ijfs, doi:10.3390/ijfs7020034_

Round 1

Reviewer 1 Report

This is an interesting paper that is well written and contributes to the literature.  There is other study that examines the predictability of the spread for recessions for China. 

I have a few comments to improve the paper

Point 1: On line 209 of the paper Table 1 should be Table 2

Point 2: In tables 2 and 3 I would like the authors to conduct a test to determine whether model 1 (with term spread) or model 2 (with user cost spread) is the preferred model

Point 3: Figure 4 is missing the shaded recession lines

Point 4: There are some misspellings in the paper (needs proof reading)

Point 5: I would like the author to discuss and cite the following paper

Ryan Mattson, “A Divisia User cost interpretation of the yield spread prediction recession,”  Journal of Risk and Financial Management, Vol. 22 (2019)

Author Response

Dear Reviewer, 

Please see attached our response to your comments. 

Thank you for your input.

Reviewer 2 Report

The Predictive Power of the User Cost Spread for Economic Recession in China and the US

This study focus on the predictive power of the yield curve slope. It provides a different leading recession indicator using the Chinese and US economy as an example. The user cost spread, based on the Divisia monetary aggregate data like the ones produced by the Center for Financial Stability, provides a better predictive ability along with a better intuitive explanation based on changes in the user cost price of holding bonds. The authors claim that these findings are the main contribution of their work. Overall, the findings of the study are quite interesting and the data analysis approach is solid. However, certain points of the study should be clearer and need some additions.

1.      The author should give more emphasis in economic meaning of the results and policy implications (e.g impact at macro-level), at least with one paragraph before the conclusion (see for instance Tsagkanos et al. 2019).

2.      The authors should provide the advantage of Probit model (for this field of research) with respect to other methodologies as Mixed Logit (Tsagkanos 2007, Train 2003)

3.      The analysis will be solid incorporating and testing theoretical hypotheses.

I think that a revised version with the abovementioned concerns could be a contribution to the literature.

Literature

Train, K. (2003): Discrete Choice Methods with Simulation, Cambridge University Press, New York.

Tsagkanos G. A., C. Siriopoulos and K. Vartholomatou (2019) “FDI and Stock Market Development: Evidence from a ‘new’ emerging market.” Journal of Economic Studies. Vol 46(1), 55-70.

Tsagkanos G. A. (2007), “A bootstrap – based minimum bias maximum simulated likelihood estimator of Mixed Logit.”, Economics Letters, 96, 282 – 286.

Author Response

Dear reviewer, 

Please see attached our responses to your comments and suggestions, and thank you very much for your effort and input to improve the quality of our paper, we really appreciate it. 

Biyan

Round 2

Reviewer 2 Report

The paper includes the suggested changes and now is ready for publication